# Agroecological Approaches in the Context of Innovation Hubs

Costin Lianu [1], Violeta-Elena Simion [2], Laura Urdes [2,3,*], Rocsana Bucea-Manea-Țoniș [4],
Irina Gabriela Radulescu [5] and Cosmin Lianu [1,6]

[1] Faculty of Economic Sciences, Spiru Haret University, 030045 Bucharest, Romania
[2] Faculty of Veterinary Medicine, Spiru Haret University, 030352 Bucharest, Romania
[3] Faculty of Management and Rural Development,
University of Agricultural Sciences and Veterinary Medicine, 011464 Bucharest, Romania
[4] Doctoral School, National University of Physical Education and Sport, 060057 Bucharest, Romania
[5] Faculty of Economic Sciences, Petroleum-Gas University of Ploiesti, 100680 Ploiesti, Romania
[6] Faculty of Economy and International Affairs, Academy of Economic Studies, 010374 Bucharest, Romania
* Correspondence: urdeslaura@gmail.com

**Abstract:** Agroecology is a sustainable alternative to agricultural science, aiming at balancing the environment-plant-animal-man complex in an equitable way. Different players in the food system across the world are engaging in the practice and promotion of agroecology. Their experience serves as input for agroecology innovation hubs, thus assisting and accelerating the adoption of agroecological practices. Based on existing experience in implementation of innovation ecosystems and living labs in Romania, the study discusses critical factors required for a successful transformation of agriculture, with the aim to fill existing research gaps on agroecological techniques. The authors are also emphasizing the role of new business models in this area. The study used an anonymous survey with Likert scale ratings, and structural equation modeling, PLS. The study results were indicative of a certain degree of enthusiasm for agroecological practice adoption, particularly among organic farmers and business owners. The chances that these practices are adopted by farmers can be enhanced provided there is a systematic exchange of knowledge among the farmers. Clusters of farmers based on community of practice could create innovation ecosystems providing this accelerates its adoption. Correlation with the economic and political processes of the country is necessary, as emphasized by the farmers interviewed during the study. Through innovation hubs, agroecology must move from the currently smaller scale to larger scale practices such as agroecosystems and agri-food systems. These forms of organization should also take due account of relevant socio-economic, cultural, and political factors.

**Keywords:** agroecological practice adoption; innovation ecosystems; innovation hubs

## 1. Introduction

Agroecology is an agriculture-related field, and a branch of general ecology. Agroecology addresses complex sustainability challenges dealing with the influences exerted by environmental factors on cultivated plants and domesticated animals (the so-called "agricultural autecology"), and the ecological research of agricultural systems (namely, "agricultural synecology") [1,2]. More than a definition, according to Food and Agriculture Organization (FAO) [3], Pigford et al. (2018) indicate that agroecology is an alternative form of agricultural science applying social and ecological concepts to agriculture management [4]. In their recently published paper, Urdes et al. (2022) showed that the approach should be sustainable because it aims at balancing the environment-plant-animal-man complex in an equitable way [5]. From a practical point of view, agroecology includes biodynamic agriculture and organic (Canada, Estonia, Unite State of America), ecological (Romania, Spain), and biological agriculture (France, Germany, Netherlands, Italy, Spain).

The rules governing the organic production and labeling of organic products are provisioned for in Regulation (EU) 2018/848 of the European Parliament and of the Council of 30 May 2018 [6].

There are new initiatives to define whether agricultural methods are agroecological or not [7]. One such initiative is represented by innovation ecosystems (IE) that refer to value creation by networking actors, through joint activities [4,8,9]. IEs are aimed at developing and commercialization of innovative products and/or services [8,9], drawing upon the former concept of business ecosystem, proposed initially by Moore (1993), cited by Gomes et al., 2016 [8]. IE differs significantly and multidimensionally from other types of non-cooperative economic concentrations of organizations in a defined territorial space [2,9,10]. Gomes et al. (2016) highlighted the differences between the business ecosystem construct and the innovation ecosystem concept, recognizing the consistency of the actor component throughout the analyzed definitions [8]. Based on these characteristics, nine types of innovation ecosystems have been identified: (i) hub-based innovation ecosystems (i.e., involve a single company assuming the ecosystem leadership [8,10]); (ii) open-source community (i.e., self-organizing and self-governing communities driven by people's needs [11–13]); (iii) research and development consortia (i.e., collaborative partnerships focused on exploiting and developing internal resources and competencies in areas where success is difficult to achieve [14]); (iv) crowdsourcing ecosystem (i.e., a business approach based on collective contributions with the aim to provide high quality solutions and to promote innovation [15]); (v) the orchestra model (i.e., a group of companies exploiting together a market opportunity based on one defined innovation structure established by one of the companies within the group [16]); (vi) creative Bazaar (i.e., a marketplace where a dominant company is looking to buy and sell innovative technologies, products, and services); (vii) Jam Central (i.e., a community of collaborating research centers aimed at developing innovative ideas, services or goods in a new or emerging field); (viii) MODification Station Model (i.e., innovation ecosystems where innovative ideas come from a community of customers who propose new uses for existing products); and (ix) family ecosystem (venture creation by family and business actors [8]).

In line with the recent initiative of the European Commission regarding the need to accelerate farming systems transition towards a green growth and circular economy through agroecology, living labs, and research [17], the paper investigates the critical success factors for a successful transformation of agriculture through innovation ecosystem, based on existing experience with implementing innovation ecosystems and living labs in Romania. The paper aims at demonstrating the existence of critical factors playing a role in the development of business models when it comes to adopting novel approaches. The paper argues that, without setting up new business models, these approaches will not be recognized by international or local markets.

## 2. Improving the Understanding of Agroecology in the European Context—Analytical Framework

The first references to agroecology were made in the education system. Wezel et al. (2009) mentioned the use of agroecology as a discipline between the years 1930–1960 [18]. Currently, education is playing an important role in agriculture practices, and agroecology must be an option in the curricula. Education for sustainable development should occupy an important place in the educational process, and it should be seen as the mission of higher education institutions [19,20]. New tools to teach this discipline are currently being addressed. A pedagogical method must be interactive and experiential. For example, the management of a mixed farm of animal and plant crops is easier for students to understand by accessing a computer game—SEGAE (SErious Game for AgroEcology learning) or in pest management (Spotted-Stop-It). Students have thus had the opportunity to correctly assess the impact of agricultural practices on some economic and social indicators, as well as the sustainability of the environment [21,22]. Many of these applications connect students and farmers, advisors or specialists in agroecological practices [23].

Later on, during 1960–1980, agroecology used to be seen as an agricultural practice focused on ecological practices applied and integrated throughout the chain, from the field crops to the farm level, and up to the entire food system [18]. However, there are clear differences between ecology and agroecology, although similarities between the two concepts are also present. Wezel et al. (2014) stated that some practices such as fertilization with organic compounds, split fertilization, reduced tillage, biological pest control, and variety selection are already integrated into European countries [24]. There are practices aimed at increasing efficiency but there are also agroecological approaches requiring a redesign based on diversification. Other practices, such as the use of intercropping, have a moderate potential to be widely implemented within the next decade [25]. Increasing the efficiency of agroecological practices in a sustainable context is not possible without innovation, even if this process takes place slowly [24,26], and is conditioned by a multitude of environmental, social, and cultural factors.

Kerr et al. (2021) [7] identified three critical issues related to sustainable agroecology practices: (i) The quantity of food required to attain FSN (food security and nutrition), focusing on whether FSN is more of an access and usage problem than a problem with availability; (ii) Could agroecological farming methods supply enough food to satisfy the world's appetite? (iii) How can the performance of food systems be measured while accounting for the numerous environmental and social externalities that are frequently disregarded in previous analyses of agricultural and food systems? Since there is not common ground on the elements that make up the concept of agroecology, a commonly accepted definition of this concept is yet to be made available. It follows that attempting to define agroecology can be challenging, but the resulting flexibility in the utilization of this approach should allow for the adaptation of agroecological practices to local needs. The transition from agriculture to agroecology generally follows several stages. These stages are: an initial increase in production efficiency by changing practices which reduce input consumption, concurrently or followed by a stage of replacing an input or some practices, and a final redesign, through which the whole system is remodeled in the sustainable direction [25]. These transformations are taking place in most countries, but the pace of transformation is variable. Environmental, social, and political factors may influence their evolution.

Currently, the concept of agroecology is perceived as a social movement, having the specific characteristics of each country, and reflecting the diversity of contexts in the world about this concept. The formation of centers of knowledge and workshops for "participation in" and "appropriation" of agricultural research with long-term experimental effects are essential in this movement. Ciaccia et al. (2019) highlighted some of the co-innovation processes that took place during the establishment of a small network of ecological farmers: contextualization of the process in the local sector, cooperation through participatory activities that make the transition from research and innovation onto the farm, defining a common language through meetings, jointly organized actions, information sharing between stakeholders [27].

Relative to the evolution of agroecology in Romania, Moudrý et al. (2018) [28] indicated that similar evolutions occurred in other countries. Romania has been a country with great agricultural potential, the particularity compared to the other EU member states being the large share of the population employed in agriculture. However, research on the transition to sustainable agriculture, and the implementation of agroecology practices are making their first steps towards implementation in the region.

Known and promoted as a discipline in the 1980s, agroecology developed as an agricultural practice and became a movement until the 2000s. An important role in the evolution was the soil bonitation of agricultural land using a system of technical indicators regarding the evaluation and interpretation of the conditions of growth and fruiting of plants and the delimitation and characterization of homogeneous ecological territories as basic units of Romanian agroecosystems.

The soil bonitation expresses the degree of favorability of the natural conditions of geography, climate, hydrology, and soil for the plants grown in Romania, and it is an indicator of appreciation of the natural and agricultural potential of the agroecosystems in Romania. It differs from one culture to another, and from one agroecosystem to another. The development of indicators and their use in evaluating the sustainability of agroecosystems (biodiversity, vulnerability, resilience, complexity, productivity, stability, and equity of its functionality, etc.) is another important step in the development of agroecology in Romania [29].

A major challenge is the persisting confusion that often arises between agroecology and ecological agriculture. In their work, Migliorini & Wezel (2017) [30] perform a rigorous analysis of the common features and the differences between ecological agriculture and agroecology. Some socio-economic principles regarding the ecological management of agri-food systems, the use of similar cultural practices such as soil fertilization, the choice of crops and their rotation, the management of pests, diseases, and weeds, the integration of culture and animal systems, and the choice of breed are identified as being approximately the same. There are differences regarding the origin and quantity of products used to combat pests, diseases, and weeds, the practices for animal management (housing, feeding, veterinary management in the prevention and treatment of diseases), technical aspects regarding food processing that are provided in ecological agriculture [30].

Numerous challenges regarding the understanding of the agroecology notion, as well as its practical, scientific, and socio-political implications are still valid. Wezel et al. (2018) suggested the following approach that would be necessary for a thorough understanding of the agroecology concept: a clear definition and common understanding of the concept, a defined role of education in promoting agroecology, investments in agroecological research, national, and international policies that support agroecological practices and transform food systems while disseminating the information about agroecology and creating meaningful alliances [31].

The High-Level Panel of Experts for Food Security and Nutrition (HLPE), the Committee on World Food Security's (CFS) science-policy interface, reached the following conclusions in 2019 regarding agroecology as a theoretically dynamic concept, as well as a body of experience:

(i) By incorporating ecological concepts into agricultural practices, it aims to transform agricultural practices, secure the sustainable use of ecosystem services and natural resources, and meet the need for socially just food systems; Technologies are applicable to all types of agricultural holdings and can play a crucial role in helping farmers use excellent agricultural practices more widely [32].

(ii) Agroecology is a transdisciplinary science that integrates various academic fields to find solutions to practical issues. It does this by collaborating with numerous stakeholders, taking into account their local knowledge and cultural values, and working in a reflective and iterative manner that encourages co-learning between researchers and practitioners as well as horizontal dissemination along the food chain.

(iii) Agroecology has developed over the past several decades to include the entirety of agriculture and food systems rather than just a concentration on fields and farms. It is not just a science but also a set of practices and a social movement. It is advised that farmers' knowledge be increased through a variety of technical training programs employing participatory methodologies, since this will encourage farmers to adopt Good Agricultural Practices (GAP) for secure pesticide usage [33].

(iv) Comprehending field-level farming techniques that prioritize recycling, maintaining soil and animal health, using little external inputs but a high level of agrobiodiversity, controlling interactions between components, and economic diversification. Since then, the emphasis has widened to incorporate processes at the landscape scale, including the more recent practices of social science and political ecology in relation to the creation of just and sustainable food systems. The likelihood of switching to organic farming is increased by the direct sales gross marketable output and by the

intensity of labor and machinery. However, the availability of family labor, farm localization, and financial resources deter the transition to an organic agricultural system [34].

(v)　An agroecological approach emphasizes the value of local knowledge and participatory processes leading to new knowledge and innovative practices through science, and the need to address social inequalities. The agroecological approach should favor the use of natural processes, limit the use of external inputs, and promote closed cycles with minimal negative externalities. This has significant ramifications for the structure of research, teaching, and extension. Gliessman (2007) outlined five stages in the shift from agroecological to more sustainable food systems. The first three are agroecological in nature and involve [35]: (i) improving input usage efficiency; (ii) switching to agroecological alternatives for conventional inputs and practices; and (iii) rebuilding the agroecosystem based on a new set of ecological processes. The final two phases, which affect the entire food system, are (iv) re-establishing a closer relationship between producers and consumers, and (v) creating a new, participatory, local, equitable, and just global food system. The latter three phases are more transformational than the first two, which are gradual. The customers' intents to choose organic food over conventional food are positively impacted by subjective norms, perceived control behavior, knowledge, health consciousness, and environmental consciousness [36].

## 3. Methodology

The current trend is the evolution of agroecology in Europe as well as in the world, as a science, agricultural practice, and social movement. In order to evaluate the degree of information and adoption of agroecology by farmers in Romania regarding agroecological practices, a questionnaire was offered for completion that included five sections: (I) Personal, social, economic, and demographic data; (II) Identification of cultivation methods; (III) Agroecological practices; (IV) Identification and characterization of the conditions/factors related to the agricultural field. (V) Importance and impact of innovation hubs in Romania. The questionnaire was answered online and aimed at identifying the knowledge of farmers regarding the methods of cultivation in an ecological system, the definition of sustainable agriculture, agricultural practices supporting biodiversity which are currently implemented, knowledge depth about the agroecology practices, on-farm specific problems, and types of regenerative agriculture practices.

The survey was anonymous, and it largely consisted of multiple-choice questions with Likert scale ratings (−2 not at all important; −1 not important; 0 neutral; 1 important; 2 extremely important). It included also open-ended questions allowing respondents to freely express their opinions. The outputs were evaluated using the statistical method of structural equation modeling using PLS, which examines concurrent interactions between latent variables, formative or reflective, even for smaller samples. This is preliminary research aiming at identifying suitable profiles for farmers and entrepreneurs in the agroecological field, to integrate them and offer specific support within innovation hubs. In this way, this exploratory study provides crucial data for further investigation. We designed our model based on 2 formative variables: Factors, Profile, and one reflective variable Practices (Table 1).

**Table 1.** Variables of the model.

| Variables | Items | Description |
|---|---|---|
| Profile | Education3 | Regarding your education, please choose one of the options (which you have already graduated from) |
| | Work4 | Regarding your work: (a) I am employed and paid, (b) I am employed and entrepreneur, (c) I am not paid, (d) None of these |
| | HA5 | The agricultural area (ha) that you have cultivated/leased is: (a) Less than 5 ha, (b) Between 5 ha and 100 ha (c) Over 100 ha |
| | Time6 | How long have you been managing your farm: |
| | EcoAgri8 | What do you mean by ecological agriculture? |
| | SustenAgri9 | What do you mean by sustainable agriculture? |
| | ImplementP11 | Are you currently implementing practices to support biodiversity? |
| Practices | Soil14 | How do you want to change the farming system you practice in the near future? On a scale of 1 to 5 I want to add: [Ground cover] |
| | Plowing14 | How do you want to change the farming system you practice in the near future? On a scale of 1 to 5 I want to add: [Plowing] |
| | Compost14 | How do you want to change the farming system you practice in the near future? On a scale of 1 to 5 I want to add: [Compost, mulch, manure] |
| | PestMng14 | How do you want to change the farming system you practice in the near future? On a scale of 1 to 5 I want to add: [Integrated pest management] |
| | Animal14 | How do you want to change the farming system you practice in the near future? On a scale from 1 to 5 I want to add: [Integrated animal husbandry] |
| | Culture14 | How do you want to change the farming system you practice in the near future? On a scale of 1 to 5 I want to add: [Diversity of cultures] |
| | Pollination14 | How do you want to change the farming system you practice in the near future? On a scale of 1 to 5 I want to add: [Pollination] |
| | Change13 | Do you want to change the farming system you practice in the near future? |
| Factors | Subsidies20 | Subsidies received |
| | CostHa21 | What are the costs per ha? |
| | IncomeHa21 | What are the incomes per ha? |
| | Profit7 | Does the farm offer you enough profits to live well? |
| | Government 22 | Are the government practices sustaining your activity? |
| | Apreciaion26 | How are you appreciated by your neighbors, in relation to the agricultural activity you carry out? |

The hypotheses of the research are:

Quantitative research was based on closed questions and mostly on continuous categorial variables.

**Hypothesis 1 (H1).** *The entrepreneurial profile of the Romanian farm manager is influenced by the conjunctural factors and thus organic farmers, and entrepreneurs are open/motivated for the adoption of agroecological practices.*

**Hypothesis 2 (H2).** *The entrepreneurial profile, and their lever of literacy in the field of agroecology influences the practices of agroecology. We may affirm that there is a relevant gap in knowledge about these practices.*

Qualitative research was based mostly on open questions.

**Hypothesis 3 (H3).** *Farmers organized in clusters are on a solid pathway towards innovation hubs.*

Taking into account the aforementioned hypotheses, the research employed SmartPls to assess the consistency through composite reliability.

## 4. Results

The farmers that cultivated more than 5 ha (Figure 1), are paid for their activity and if there are entrepreneurs in this field who know very well what the ecological agriculture is, they have a stronger entrepreneurial profile, are rather young men (26–45 years old) managing their own farm for less than 10 years (Figure 2). They do not currently implement practices to support biodiversity, they accuse specific problems they have encountered on the farm (example) and do not want to change the farming system in their practice in the near future, maybe because they learned how to practice agriculture from their family and acquired their skills, knowledge, and experience without attending specialized studies.

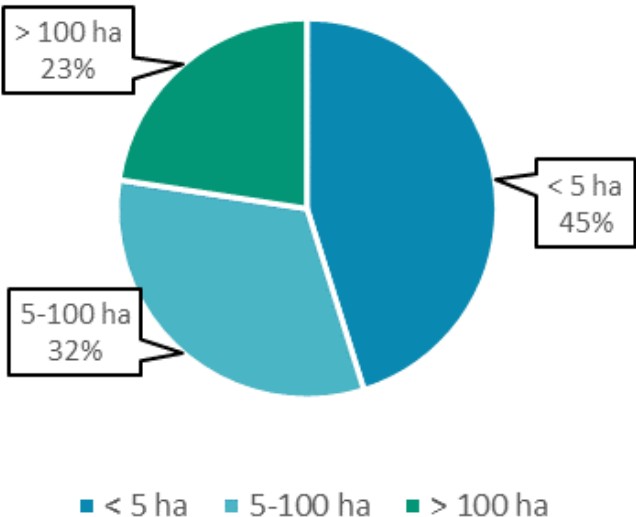

**Figure 1.** Agricultural area (ha) cultivated/leased.

Currently, they are only implementing biodiversity support practices to a small extent (Figure 3), (1—never, 5—very frequently) maybe because they have learned to practice farming from their family and acquired their skills, knowledge, and experience without going through specialized studies, and they blame specific problems they encountered on the farm (for example, problems with human resources, lack of employee skills, lack of technological resources, etc.).

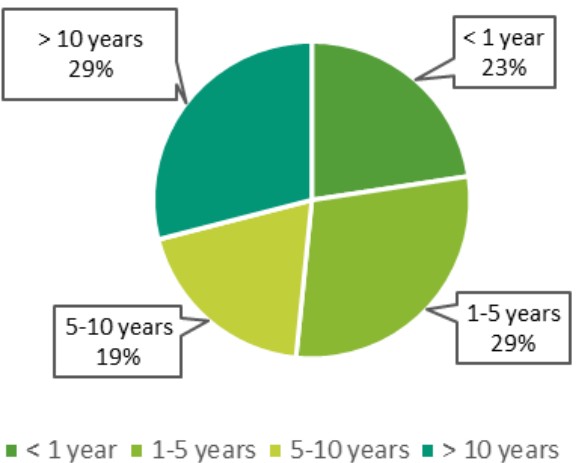

**Figure 2.** Period of farm administration/employment.

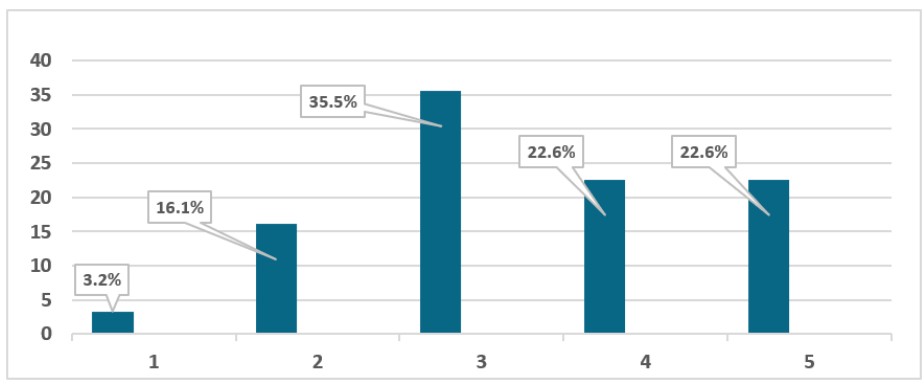

**Figure 3.** Are you currently implementing practices that support biodiversity?

At the same time, it was important to know what the farmers understand by sustainable agriculture (Figure 4). A significant percentage of the respondents (46%) appreciated soil protection and biodiversity.

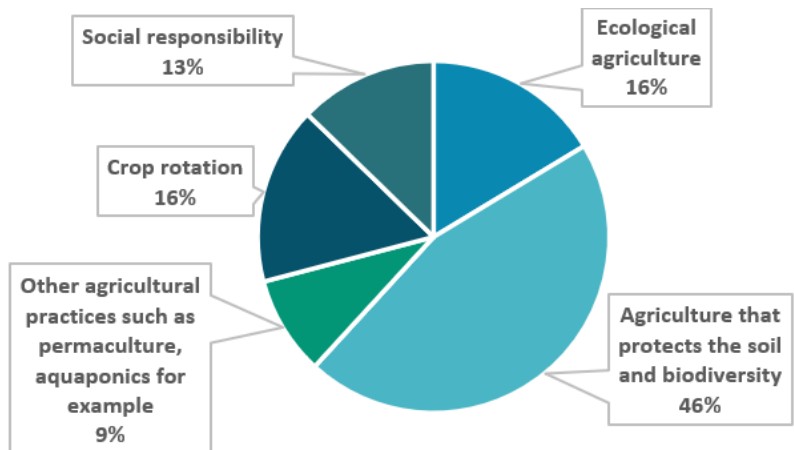

**Figure 4.** What do you understand by sustainable agriculture?

However, they are willing to change the farming system they currently practice, which can be a good start in supporting the transition to agroecology (Figure 5).

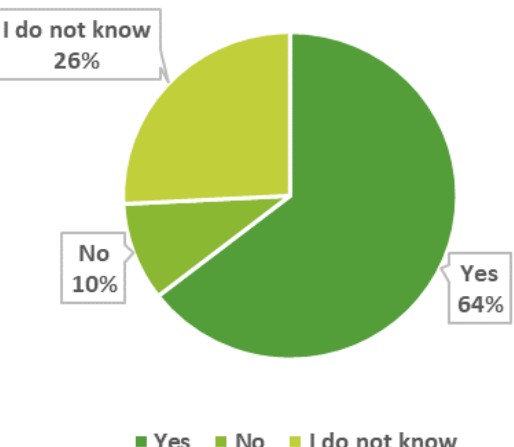

**Figure 5.** Do you want to change the farming system you practice in the near future?

Each variable of our model is composed by many items. For example, the formative variable Factors is made of six items evaluating the influence of the context on the entrepreneurial profile in agroecology filed. The item with the highest weight was CostHa21. The high loading factor (LF = 0.700) of this variable emphasizes that the farmers consider that the cost per Ha is a very important factor that influences the agroecological practices. Very related to cost is the profit. The entrepreneurs consider that the profit is the second important factor that influences these practices. The Profit17 variable has a loading factor of 0.413. Other factors that influence the entrepreneurial profile are Susidies20 (LF = 0.312), IncomeHa21 (LF = 0.176), Appreciation26 (LF = 0.120), and Gouvernment22 (0.097). Thus, we may affirm that the farmers are aware of the subsidies available in this field and use them in their activity. They are motivated by the fact that the neighbors appreciate the agricultural activity carried out by them and by the government support, in a small measure (Figure 6).

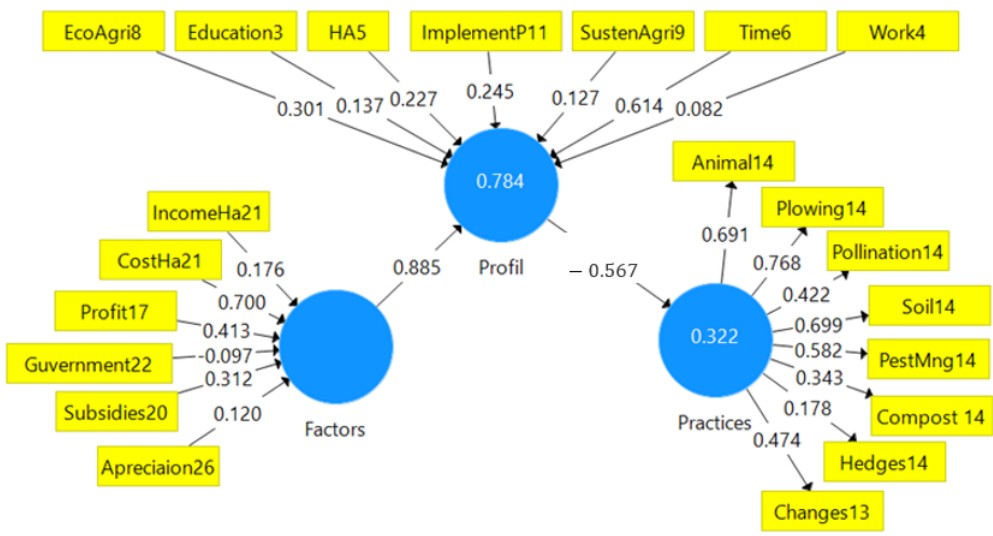

**Figure 6.** Cronbach Alpha Analysis and Path Coefficients.

The Profile variable is formed by seven items and is rather positively influenced by the Time6 (LF = 0.614), EcoAgri8 (LF = 0.301), HA5 (LF = 0.227), and ImplementP11 (LF = 0.245). Other factors that form the user profile are Education (LF = 0.137), SustenAgri9 (LF = 0.127), and Work4 (LF = 0.082). The farmers that work in the field by more than five years are keener to implement Ecological and Sustainable agriculture and sustain biodiversity. The

data show that some of them are entrepreneurs in this field and make a profit from this activity (Figure 6).

The reflective variable Practices made of eight items is rather influenced by the way of changing their farming system into the near future implementing Pwowing14 (LF = 0.768), Animal14 (LF = 0.691), Soil14 (LF = 0.699), PestMng14 (LF = 0.582), Pollination (LF = 0.442), Compost14 (LF = 0.343), Hedges14 (LF = 0.178), and Changes13 (LF = 0.474). Most of the farmers are already prepared to implement changes in their practice such as ground cover, plowing, compost, mulch, manure, Integrated pest management, Integrated animal husbandry, pollination, and diversity of cultures.

For our reflective variables Practices, the value (CR = 0.757) is greater than 0.7, the minimal threshold acceptable (Table 2). For the other variables CR and CA are not calculated, because they are formative. Additionally, the Path coefficient Factors → Profile (0.885) is very high (Figure 5). We can say with certainty the factors chosen into the model plays a significant role in the entrepreneurial profile. The Path coefficient Profile → Practices (−0.567) has a negative influence. The entrepreneur has to be educated regarding the ago-environmental practices through an innovation hub.

**Table 2.** Validation Steps/tests.

| Reflexive/Formative Construct | Composite Reliability | Cronbach Alpha | rho_A | R Square | Path Coefficients | |
|---|---|---|---|---|---|---|
| | (>0.7) | (>0.7) | (>0.5) | (>0.5) | | |
| Pracitices | 0.757 | 0.794 | 0.808 | 0.522 | Factors-Profile | 0.885 |
| Factors | | | 1 | | Profile-Practices | −0.567 |
| Profile | | | 1 | 0.784 | | |

The Cronbach's Alpha Analysis shows that the survey questions were very well chosen—the factors (sub-indicators) that influence Practices (CA = 0.753), represent the analysis because they all have good values.

A strong positive correlation is observed between Factors and Profile (0.885), a medium negative correlation is presented between Profile and Practices (−0.567). The Chi-Square for the estimated model (292.366) is greater than the Chi-Square for the saturated model (291.811). Thus, we may affirm that our model fits and that H1 and H2 hypotheses are accepted (Table 3).

**Table 3.** Variable correlation and model fit.

| Latent Variable Correlation | | | |
|---|---|---|---|
| Variable | Factors | Practices | Profile |
| Factors | 1 | | |
| Practices | −0.455 | 1 | |
| Profile | 0.885 | −0.567 | 1 |
| **Fit Summary** | | | |
| Saturated Model | | Estimated Model | |
| 291.811 | | 292.366 | |

The Variance Inflation Factor (VIF) of each construct was determined by SmartPL's software to assess the relevance of variables. Table 4 provides a summary of the findings. Since no VIF values are greater than five, there is no multicollinearity between the variables.

**Table 4.** Collinearity statistics.

| Variable | VIF | Variable | VIF | Variable | VIF | Variable | VIF |
|---|---|---|---|---|---|---|---|
| Appreciation26 | 1.447 | Education3 | 1.200 | ImplementP11 | 1.297 | Practices10 | 2.311 |
| animal14 | 2.652 | EnvGrants23 | 3.645 | Knowledge7 | 1.172 | Problem12 | 1.748 |
| Change13 | 1.506 | Facil16 | 1.578 | Law23 | 2.025 | Soil14 | 2.497 |
| Compost14 | 2.465 | Gender2 | 1.471 | MetTech25 | 1.082 | SustenAgri9 | 1.123 |
| CostHa21 | 1.538 | Hedges14 | 3.211 | OK23 | 3.877 | Time6 | 1.362 |
| EcoAgri8 | 1.161 | Government22 | 1.493 | PestMng14 | 2.102 | Work4 | 1.166 |
| HA5 | 1.205 | HA5 | 1.439 | Pollinatio14 | 2.069 | Subsidies20 | 1.182 |

The *t* Test Statistics are representative and the *p* values for all three SEM regressions are less than the 0.05 threshold, showing again that our models were well designed (Table 5). The bootstrapping value of two-tailed *t* tests was greater than 1.96 [37,38].

**Table 5.** The *t* Test Statistics and *p* Values of the Bootstrapping Analysis.

| | Original Sample (O) | Sample Mean (M) | Standard Deviation (STDEV) | *t* Test Statistics (|O/STDEV|) | *p* Values |
|---|---|---|---|---|---|
| Factors → Profile | 0.885 | 0.903 | 0.087 | 10.159 | 0.000 |
| Profile → Practices | −0.567 | −0.724 | 0.281 | 2.021 | 0.044 |

The steps presented in Tables 2–5 empower us to assume that the indicators of the constructs correlate and are appropriate for the model and the model is representative and fit.

## 5. Discussions

Many areas are still underexplored in agroecological research. Such areas are the economic performance of agroecological practices and their adoption of efficient business models compared to readily available alternatives; connecting agroecology means with public policy instruments; the economic and social impact of adopting agroecological approaches; the role of innovation ecosystems and the extent to which agroecological practices increase resilience to climate change threats.

The holistic approach in this field is the only sustainable option to face the complex challenges in agriculture, from production to consumption. For a better future, FAO elaborated the 2030 Agenda for Sustainable Development to ensure food security and safety, aligned with human rights and focused on ten targets, including food security, nutrition, and health, climate change resilience, and biodiversity. However, the approach to agroecological practices depends on a series of local, governmental, economic, and cultural factors. Transition to agriculture systems that are sustainable, i.e., preserve soil, water, plant and animal genetic resources, being at the same time socio-economically appropriate, viable, and acceptable, imply cross-sector and facilitation of transboundary innovation, as well as a multi-level perspective on innovation ecosystems [28]. The innovation Ecosystems approach conceptualizes the need for alternative forms of agriculture, which draws upon the potential for the development of circular economies [39]. Transition to sustainability requires that the spaces allowing one to innovate, and institutional entities supporting the transition, known as "innovation niches", advance innovation by working across scales [40], while acknowledging that characteristics of the outcomes at one scale are shaped by the flows and interactions occurring in other scales [41]. While agroecology has long sought to integrate multiple scales to advance innovation and scaling of new agroecological systems [42], the Innovation Ecosystems approach considers emergent effects due to feedback loops between scales [4], and it is more explicit on the need to continuously move and adaptively engage with different scales [43].

In order to improve these ecosystems by making them operate more effectively and efficiently, a variety of organizations and institutions, including local authorities, universities, governments, corporations, investors, entrepreneurs, technological brokers, media,

startup accelerators, and other players, are included in innovation ecosystems. Such ecosystems may take the form of industrial districts, living labs, innovation centers, clusters or mega-clusters (cluster consortia), or other kinds of actor interaction. Communities, farmers, processors, certifying agencies, and other organizations use innovation for sustainable food-producing agroecological practices to improve or provide new goods and services in the design, manufacturing, or recycling of goods and services, as well as changes in the institutional context. Changes in behaviors, social mores, economic conditions, and institutional structures contribute to the emergence of potentially disruptive new food production, processing, distribution, and consumption networks. A sustainable city-region food system may be established by scaling out agroecology, which boosts the resilience of the urban food environment [44,45].

In Romania, organic farming is perceived as the "most effective and environmentally friendly solution to the growing pressure on land resources as a result of population growth and urban demand for goods and services" (Popovici et al., 2018, 2020). There are some regional disparities in the dynamics of organic areas, with organic producers located in the mountain-plateau-hilly region in the center, north and north-east of the country, where livestock prevails, and in the plain regions of the west, south, and south-east, dominated by crop producing [16,46]. The use of spatial clustering is strongly dependent on the local environmental conditions. As Andrei et al. (2015) indicated in their study, conventional use of the land leads to environmental damage and degradation of ecosystems, and the economic efficiency is slightly higher in organic systems compared to conventional systems [47]. Farmers converted some of their land to organic farming, as shown by the statistics of accelerated growth. Other authors are also indicating the importance of the organic sector as a catalyzer for agroecology development in Romania [48,49].

Innovation ecosystems in agroecology are based on innovation hubs formed by: corporate R&D Labs, Co-Working Spaces, Innovation labs, and living labs. For the purpose of creating training and consulting programs, and to improve agricultural resilience, a better knowledge of farmers' attitudes is required [50]. In the case of clusters, for instance, the territorial dimension or the sector orientation of the member companies is more obvious and closer to a business alliance model. "Living labs", defined as an arena for supporting experimentation in a natural setting with various stakeholders [51] on the other hand, are a relatively new approach to clusters to action-oriented research, which uses novel technologies in actual setups, with the intent to foster learning through communication between participants. Social and organizational features of living labs are of utmost importance, to the detriment of business alliance orientation in clusters, but both are fully engaging, among other actors, R&D institutions, and universities. IE may address different markets and the development of products. Early adopters focus on diversification, mixed farming, intercropping, cultivar mixtures, habitat management methods for crop-associated biodiversity, biological pest control, improving soil structure and health, biological nitrogen fixation, and nutrient, energy, and waste recycling [52].

The study aimed at identifying the best approach that we can have in Romania, for the transition to the agricultural innovation/innovation ecosystems and how we ensure this transition. Thus, two hypotheses were verified: the entrepreneurial profile of the farm manager in Romania is influenced by the conjunctural factors that motivate them to adopt agroecological practices in their activity and the way they learned to practice agriculture—from the family and by accessing specialized studies that influence agroecology practices.

The interviewed farmers in our study believe that an important factor for the application in agroecology is the cost per Ha. Implicitly, related to this is the profit. Entrepreneurial farmers consider profit to be the second important factor influencing these practices. A real support for them is the subsidies they receive in this field.

Farmers are less motivated by the fact that neighbors appreciate their activity in the agricultural field, an important factor being the lack of associations in agriculture as well as government support. They recognize the importance of subsidies and believe that it is

necessary for the financial value to increase, but also for the national policies that must support this field.

Farmers from the marshes along the Danube created the Bio Danubius Hub participating in this study. This is an innovation hub created with authorities, local business environments and the university entrepreneurial center USH ProBusiness, which participates in the sustainable development of the region, based on the principles of agroecology and bioeconomy. An objective of this cluster is the elaboration of legislative proposals with impact on the agricultural field and with future political implications in the agroecological field in Romania. The entrepreneurial component and the academic and research experience of the Bio Danubius Hub can generate business models that are based on results in the context of the European Union and that can be recognized in the local and international markets.

Experience in agriculture has an important role in implementing changes towards the transition to agroecology. Farmers with more than five years of experience are more open and accepting of the implementation of ecological and sustainable agriculture and supporting biodiversity. In addition, some farmers who are also entrepreneurs obtain profit from this activity.

Most of the farmers interviewed already practice changes in their practice, such as the use of compost and manure, integrated pest management, pollination, crop diversity, and the use of hedges and forestry. This aspect is important in supporting the transition toward sustainable agriculture.

## 6. Conclusions

The main hypotheses of our study are confirmed. Organic farmers in Romania, as well as others, have embraced agroecological principles. Farmers' willingness to embrace the strictness of certifications that are already in the ecological field is correlated with the benefit of the organic movement toward adopting such techniques. In order to limit the usage of bought inputs such as fossil fuels and agrochemicals, and to develop more diversified, resilient, and productive agroecosystems, they are more inclined to harness, sustain, and increase biological and ecological processes in agricultural production. The model also shows that factors such as grants, government policies, legislation, knowledge or local cohesion in clusters are important. At the same time, the model shows a higher propensity to change among more informed farmers, able to exchange information with other framers or consultants in systematic management, as with the innovation hubs established in this field. The lack of identification of a valuable business model constitutes an obstacle which is easier to overcome for organic farmers.

Early adopters are paying attention to diversification, mixed farming, intercropping, cultivar mixtures, habitat management methods for crop-associated biodiversity, biological pest control, bettering soil structure and health, biological nitrogen fixation, recycling of nutrients, energy, and waste [52].

Farmers in the marshes along the Danube River created the Bio Danubius Hub. Depending on the extent to which agroecological principles are followed locally, they are in the process of identifying categorized indicators on the inorganic-agroecological spectrum.

The results of this research indicated clearly that the proposed study is timely and pertinent in terms of the knowledge gap. Some of our findings are:

- reliance on ecological processes is not clear from the point of view of costs and benefits, and as a business model.
- local adaption and control of a system's approach embracing management of interactions among components rather than focusing only on specific technologies is not clearly understood.
- understanding it as a social movement associated with agroecology is not obvious, and it requires greater efforts to initiate a widespread change of agriculture and food systems.

- agroecology is not understood as an interinstitutional political framework under which many social movements and peasant organizations around the world assert their collective rights, and advocate for a diversity of locally adapted agriculture and food systems, mainly practiced by small-scale food producers.
- there is a need for a strong connection to be made between agroecology, the right to food and food sovereignty, but it is not clear how to connect these areas.
- agroecology is seen as a political struggle, requiring people to challenge and transform governance structures and society at large, but it is not clear how and at what costs.

There are different challenges identified for the development of agroecology in Europe, as presented by Wezel et al. (2018) among which we list education and training, research funding, policies, consumer awareness, etc. [31]. Communication and the creation of alliances in the field of agroecology, through Innovation Hubs, contribute to the implementation of the concept starting from the local level, as a solid basis for support at the national and European level. A sustainable development must be done through education and innovation, and in this context, the Innovation Hubs are the core of the actions. Future activities that will be implemented in the short term through the Bio Danubius Hub include the introduction of the discipline of Agroecology in the curriculum of specialized faculties, such as economics and veterinary medicine; the creation of business models that support farmers so that work in agriculture becomes profitable; the elaboration of some legislative proposals in the field of agroecology that will be supported by the innovation hubs, as policy recommendations (stronger financial support for farmers, for example).

## 7. Limitations of the Study

Agriculture is the main scale at which the concept of agroecology is applied in Romania. In this context, the development of the conceptualization of agroecology in Romania takes place in parallel as a science and some agricultural practices and is just at the beginning, as a social movement. Innovation centers play an important role in its implementation. Correlation with the economic and political processes of the country is necessary, as emphasized by the farmers interviewed during the study.

Future research related to this study can be related to the presentation of business models with impact in the practice of agroecology. Regionalization and nationalization of these concepts would also be necessary. Through innovation hubs, agroecology must move from practices at the scale of agroecosystems and agri-food systems, including economic, social, cultural and political aspects.

**Author Contributions:** Conceptualization, C.L. (Costin Lianu), V.-E.S., L.U., I.G.R. and C.L. (Cosmin Lianu); methodology, C.L. (Costin Lianu), V.-E.S., R.B.-M.-Ț. and I.G.R.; software, C.L. (Costin Lianu), V.-E.S., R.B.-M.-Ț. and C.L. (Cosmin Lianu); validation, C.L. (Costin Lianu), V.-E.S. and I.G.R.; formal analysis, C.L. (Costin Lianu), V.-E.S., R.B.-M.-Ț. and L.U.; investigation, C.L. (Costin Lianu), V.-E.S., R.B.-M.-Ț. and I.G.R.; resources, C.L. (Costin Lianu), V.-E.S., L.U. and R.B.-M.-Ț.; data curation, C.L. (Costin Lianu), V.-E.S., I.G.R. and C.L. (Cosmin Lianu); writing—original draft preparation, C.L. (Costin Lianu), V.-E.S., L.U. and R.B.-M.-Ț.; writing—review and editing, C.L. (Costin Lianu), V.-E.S., L.U., R.B.-M.-Ț., I.G.R. and C.L. (Cosmin Lianu); visualization, C.L. (Costin Lianu), V.-E.S., R.B.-M.-Ț. and C.L. (Cosmin Lianu); supervision. C.L. (Costin Lianu) and V.-E.S.; project administration, C.L. (Costin Lianu) and V.-E.S. All authors have read and agreed to the published version of the manuscript.

**Funding:** This research received no external funding.

**Institutional Review Board Statement:** The study was conducted in accordance with the Declaration of Helsinki and approved by the Institutional Review Board (or Ethics Committee) of Faculty of Medicine, Spiru Haret University, which was the study lead.

**Informed Consent Statement:** Informed consent was obtained from all subjects involved in the study.

**Data Availability Statement:** Not applicable.

**Conflicts of Interest:** The authors declare no conflict of interest.

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
