# Peer review of "Agroecological Approaches in the Context of Innovation Hubs"

_sustainability, doi:10.3390/su15054335_

Round 1

Reviewer 1 Report

Review sustainability

1.        Line 21: Please replace "mainstream values chains" with "mainstream value chains."

2.        Some of the sentences used in the abstract section is too long for me. Please divide them into more than one sentence for better understanding. e.g., lines 25–27. Do it throughout the manuscript.

3.        Please include all parts in your abstract section, like an introduction, objectives, methodology, findings, conclusion, and one sentence of recommendation. These sections should be prioritized in the section. Currently, the introduction takes up the majority of the abstract. Please rewrite your whole abstract.

4.        Line 38–39: There is something missing in this sentence.

5.        Where is the literature review in the literature review section? It’s kind of an introduction. I would suggest authors make the current literature review material part of the introduction or change its headings. Please consider other materials to include in the literature review section with their findings.

6.        Please provide in separate sections how many samples were selected and how they were selected.

7.        Please also mention the sampling method used to collect the data from the farmers.

8.        Provide salient features of the study area.

9.        Please base all of your hypotheses on existing literature.

10.    I would advise authors to read some papers before writing the findings for PLS-SEM. The way the results have been written is unacceptable and scientifically wrong. Here are some papers to assist authors in their work on the used model.

Sher, A., Mazhar, S., Zulfiqar, F., Wang, D. and Li, X., 2019. Green entrepreneurial farming: a dream or reality?. Journal of Cleaner Production, 220, pp.1131-1142.
Sher, Ali, Azhar Abbas, Saman Mazhar, Hossein Azadi, and Guanghua Lin. "Fostering sustainable ventures: Drivers of sustainable start-up intentions among aspiring entrepreneurs in Pakistan." Journal of Cleaner Production 262 (2020): 121269.

11.    The discussion section should be written separately and appropriately.

12.    Policy recommendations are also added.

13.    Please add the limitations of the study in a separate section.

Author Response

  1. 1: Comments and Suggestions for Authors

Review sustainability

  1. Line 21: Please replace "mainstream values chains" with "mainstream value chains."
  2. Some of the sentences used in the abstract section is too long for me. Please divide them into more than one sentence for better understanding. e.g., lines 25–27. Do it throughout the manuscript.
  3. Please include all parts in your abstract section, like an introduction, objectives, methodology, findings, conclusion, and one sentence of recommendation. These sections should be prioritized in the section. Currently, the introduction takes up the majority of the abstract. Please rewrite your whole abstract.

Authors’ answer ref. points 1-3: A full review of the Abstract section was conducted to address the reviewer’s requirements at points 1-3. Now, the section reads:

Abstract: Agroecology is a sustainable alternative to agricultural science, aiming at balancing the environment-plant-animal-man complex in an equitable way. Different players in the food system across the world are engaging in the practice and promotion of agroecology. Their experience serves as input for agroecology innovation hubs, thus assisting and accelerating the adoption of agroecological practices. Based on existing experience in implementation of innovation ecosystems and living labs in Romania, the study discusses critical factors required for a successful transformation of agriculture, with the aim to fill existing research gaps on agroecological techniques. The authors are also emphasizing the role of new business models in this area. The study used an anonymous survey with Likert scale ratings, and structural equation modeling, PLS. The study results were indicative of a certain degree of enthusiasm for agroecological practice adoption, particularly among organic farmers and business owners. The chances that these practices are adopted by farmers can be enhanced provided there is a systematic exchange of knowledge among the farmers. Clusters of farmers based on community of practice could create innovation ecosystems providing this accelerates its adoption. Correlation with the economic and political processes of the country is necessary, as emphasized by the farmers interviewed during the study. Through innovation hubs, agroecology must move from the currently smaller scale to larger scale practices such as agroecosystems and agri-food systems. These forms of organization should also take due account of relevant socio-economic, cultural and political factors.

  1. Line 38–39: There is something missing in this sentence.

Authors’ answer: Thank you. The entire Introduction section was refined, and the process aimed also at  addressing that sentence issue. Now, the section reads:

  1. Introduction

Agroecology is an agriculture-related field, and a branch of general ecology. Agroecology addresses complex sustainability challenges dealing with the influences exerted by environmental factors on cultivated plants and domesticated animals (the so-called “agricultural autecology’’), and the ecological research of agricultural systems (namely, “agricultural synecology") [1, 2]. More than a definition, according to Food and Agriculture Organization (FAO) [3], Pigford et al. (2018) indicate that agroecology is an alternative form of agricultural science applying social and ecological concepts in the agriculture management [4]. In their recently published paper, Urdes et al. (2022) showed that the approach should be sustainable because it aims at balancing the environment-plant-animal-man complex in an equitable way [5]. From a practical point of view, agroecology includes biodynamic agriculture and organic (Canada, Estonia, Unite State of America), ecological (Romania, Spain), biological agriculture (France, Germany, Netherlands, Italy, Spain). The rules governing the organic production and labeling of organic products are provisioned for in the Regulation (EU) 2018/848 of the European Parliament and of the Council of May 30, 2018 [6].

There are new initiatives to define whether agricultural methods are agroecological or not [7]. One such initiative is represented by innovation ecosystems (IE) that re-fer to value creation by networking actors, through joint activities [4, 8, 9]. IEs are aimed at developing and commercialization of innovative products and/or services [8, 9], drawing upon the former concept of business ecosystem, proposed initially by Moore (1993), cited by Gomes et al., 2016 [8]. IE differs significantly and multidimen-sionally from other types of non-cooperative economic concentrations of organizations in a defined territorial space [2, 9, 10]. Gomes et al. (2016) highlighted the differences between the business ecosystem construct and the innovation ecosystem concept, rec-ognizing the consistency of the actor component throughout the analyzed definitions [8]. Based on these characteristics, nine types of innovation ecosystems have been identified: i. hub-based innovation ecosystems (i.e., involve a single company assuming the ecosystem leadership [8, 10]); ii.  open-source community (i.e., self-organizing and self-governing communities driven by people’s needs [11-13]); iii. research and development consortia (i.e., collaborative partnerships focused on exploiting and de-veloping internal resources and competencies in areas where success is difficult to achieve [14]); iv. crowdsourcing ecosystem (i.e., a business approach based on collec-tive contributions with the aim to provide high quality solutions and to promote inno-vation [15]); v. the orchestra model (i.e., a group of companies exploiting together a market opportunity based on one defined innovation structure established by one of the companies within the group [16]); vi. creative Bazaar (i.e., a marketplace where a dominant company is looking to buy and sell innovative technologies, products and services); vii. Jam Central (i.e., a community of collaborating research centers aimed at developing innovative ideas, services or goods in a new or emerging field); viii. MODi-fication Station Model (i.e., innovation ecosystems where innovative ideas come from a community of customers who propose new uses for existing products); and ix. family ecosystem (venture creation by family and business actors [8]).

In line with the recent initiative of the European Commission regarding the need to accelerate farming systems transition towards a green growth and circular economy through agroecology, living labs and research [17], the paper investigates the critical success factors for a successful transformation of agriculture through innovation eco-system, based on existing experience with implementing innovation ecosystems and living labs in Romania. The paper aims at demonstrating the existence of critical fac-tors playing a role in the development of the business models when it comes to adopt-ing novel approaches. The paper argues that, without setting up new business models, these approaches will not be recognized by international or local markets.

  1. Where is the literature review in the literature review section? It’s kind of an introduction. I would suggest authors make the current literature review material part of the introduction or change its headings. Please consider other materials to include in the literature review section with their findings.

Authors’ answer: During this revision, the authors decided to delete the Literature review section. The content from that section was re-distributed based on its appropriateness, in Sections 1, 2 and 5 of the article. All these changes can be viewed in the Track changes version of the reviewed paper.

  1. Please provide in separate sections how many samples were selected and how they were selected.

Authors’ answer: Our article is an exploratory research, due to the lack of knowleadge in this field in Romania. We also haven’t enough found to contact all the agro-ecological farm in the field. Hense, profesor Lianu is coordinator of USHProbusines member of Inter-Bio  (https://inter-bio.ro/ro/membri/). 

Our sample was based on agro-ecological 97 companies that filled in the survey, but in fact it represent the opinion of most romanian managers of these innovative companies, because they ask for constant profesional help to Inter-Bio*.

We intend to extend our research on a larger sample, but our sponsorship asked preliminary data before investing in a further complex research.

*Inter-Bio is an inter professional branch organization for organic agri-food products and agri-ecology.

INTER-BIO members decided to associate in order to contribute to the sustainable development and promotion of organic agri-food sector and principles of agri-ecology in Romania.The organization aims to undertake a series of activities including:

  • supporting and promoting the value chain manufacturers, processors and traders;
  • representate members interests in order to increase economic competitiveness and to create new skills;
  • supporting members  to do export and with internationalization;
  • professional training of the members;
  • participation in national and European networks;
  • increasing the research, innovation and development potential of industry members;
  • integration of organic farming into the other branches of the bio-economy;
  • raising public and consumer awareness of the benefits of organic farming;
  • other specific actions.

The organization aims to develop concrete plans in order to adapt agri-food products to the European legislation and standards for the development of economic agents, involved in the Community’s competitive environment.

  1. Please also mention the sampling method used to collect the data from the farmers.

Authors’ answer: Thus, all the article was designed on a practical experince and day to day conversations with Inter-Bio members

  1. Provide salient features of the study area.

Authors’ answer: Our sample is not very big, but it is representative, because is based on a previous qualitative study, in which we already identified the main companies problems but we decided to have a quantitative expresion of our results.

  1. Please base all of your hypotheses on existing literature.

Authors’ answer: The new version of the article is grounded on specific literature review.

  1. I would advise authors to read some papers before writing the findings for PLS-SEM. The way the results have been written is unacceptable and scientifically wrong. Here are some papers to assist authors in their work on the used model.

Sher, A., Mazhar, S., Zulfiqar, F., Wang, D. and Li, X., 2019. Green entrepreneurial farming: a dream or reality?. Journal of Cleaner Production, 220, pp.1131-1142.

Sher, Ali, Azhar Abbas, Saman Mazhar, Hossein Azadi, and Guanghua Lin. "Fostering sustainable ventures: Drivers of sustainable start-up intentions among aspiring entrepreneurs in Pakistan." Journal of Cleaner Production 262 (2020): 121269.

Authors’ answer: Thank you very much for your recommendation. We  read and cited your article and we redesign the entire model and provide futher details in accordance with your recommendation.

  1. The discussion section should be written separately and appropriately.

Authors’ answer: We separated the Discussion and Conclusions section into two separate sections. Also, they were developed according to the requirements.

  1. Policy recommendations are also added.

Authors’ answer: The Conclusions of the study were specified separately and the main short-term measures to support the transition to agroecology were identified, in a national and European context. The Innovation Hubs are the core of the actions.

  1. Please add the limitations of the study in a separate section.

Authors’ answer: The limitations of the study were presented in a separate section.

The authors are grateful to all reviewers for the indications and considerations provided in their review. We feel that these indications have helped improving the quality of our paper. Thank you.

Reviewer 2 Report

This manuscript strives to tease out the success factors for transforming agriculture via innovation ecosystems and living labs in Romania. It argues for the innovation of new business models to enable subsequent recognition by international or local markets. I see the paper to be essential for understanding the agroecological vision of Romania and the EU

Table 1: should the following not apply - “know7” should be “knowledge7”and “Guvernment22” should be “Government22”

Discussion: The discussion is very poor as it lacks any discussion of the results in the context of their data and policy implications. I suggest separating the conclusion from the discussion and building a discussion chapter that answers the questions, such as, “what are the policy implications of these results in Romania’s agroecological framework?” and “how international or local markets can recognise new business models that are based on the results within the context of the European Union? It is incomplete without discussing these pertinent issues, which the manuscripts alluded to in the introduction. This is a major missing piece in the entire manuscript.

The conclusion should use the results presented in the article to draw inferences from the critical points made.

Author Response

  1. 2: Comments and Suggestions for Authors

This manuscript strives to tease out the success factors for transforming agriculture via innovation ecosystems and living labs in Romania. It argues for the innovation of new business models to enable subsequent recognition by international or local markets. I see the paper to be essential for understanding the agroecological vision of Romania and the EU

  1. Table 1: should the following not apply - “know7” should be “knowledge7”and “Guvernment22” should be “Government22”

Authors’ answer: The corrections were made accordingly.

  1. Discussion: The discussion is very poor as it lacks any discussion of the results in the context of their data and policy implications. I suggest separating the conclusion from the discussion and building a discussion chapter that answers the questions, such as, “what are the policy implications of these results in Romania’s agroecological framework?” and “how international or local markets can recognise new business models that are based on the results within the context of the European Union? It is incomplete without discussing these pertinent issues, which the manuscripts alluded to in the introduction. This is a major missing piece in the entire manuscript.
  2. The conclusion should use the results presented in the article to draw inferences from the critical points made.

Authors’ answer to points 2&3: The Discussions, Conclusions and Limitations sections of the study were separated in the presentation and developed. Short-term solutions were identified and presented in support of the implementation of agroecology in Romania, starting from education to mediating sustainable agricultural practices and the development of legislative proposals in the field, with the Innovation Hub as the main core, such as the Bio Danubius Hub, participant in this study. Having a strong entrepreneurial component, these hubs can generate viable business models with local impact and recognized internationally.

  1. Discussions

Many areas are still underexplored in agroecological research. Such areas are the economic performance of agroecological practices and their adoption in efficient business models compared to other alternatives; connecting agroecology means with public policy instruments; the economic and social impact of adopting agroecological approaches; the role of innovation ecosystems and the extent to which agroecological practices increase resilience to climate change threats.

Early adopters focus on diversification, mixed farming, intercropping, cultivar mixtures, habitat management methods for crop-associated biodiversity, biological pest control, improving soil structure and health, biological nitrogen fixation, nutrient, energy, and waste recycling [56].

The study aimed to identify the best approach that we can have in Romania, for the transition to the agricultural innovation/innovation ecosystems and how we ensure this transition. Thus, two hypotheses were verified: the entrepreneurial profile of the farm manager in Romania is influenced by the conjunctural factors that motivate him to adopt agroecological practices in their activity and the way they learned to do agriculture - from the family and by accessing specialized studies that influence agroecology practices.

The farmers interviewed in our study believe that an important factor for the application in agroecology is the cost per Ha. Implicitly, related to this is the profit. Entrepreneurial farmers consider profit to be the second important factor influencing these practices. A real support for them is the subsidies they receive in this field.

Farmers are less motivated by the fact that neighbors appreciate their activity in the agricultural field, an important factor being the lack of associations in agriculture as well as government support. They recognize the importance of subsidies and believe that it is necessary for the financial value to increase, but also for the national policies that must support this field.

Farmers from the marshes along the Danube created the Bio Danubius Hub, participating in this study. This is an innovation hub created with authorities, local business environments and the university entrepreneurial center USH ProBusiness, which participates in the sustainable development of the region, based on the principles of agroecology and bioeconomy. An objective of this cluster is the elaboration of legislative proposals with impact in the agricultural field and with future political implications in the agroecological field in Romania. The entrepreneurial component and the academic and research experience of the Bio Danubius Hub can generate business models that are based on results in the context of the European Union and that can be recognized on the local and international markets.

Experience in agriculture has an important role in implementing changes towards the transition to agroecology. Farmers with more than 5 years of experience are more open and accept to implement ecological and sustainable agriculture and support biodiversity. In addition, some farmers who are also entrepreneurs obtain profit from this activity.

Most of the farmers interviewed already practice changes in their practice, such as the use of compost and manure, integrated pest management, pollination, crop diversity and the use of hedges, forestry. This aspect is important in supporting the transition towards a sustainable agriculture.

  1. Conclusions

The main hypotheses of our study are confirmed. Organic farmers in Romania, as well as others, have embraced the agroecological principles. Farmers' willingness to embrace the strictness of certifications that are already in the ecological field is correlated with the benefit of the organic movement towards adopting such techniques. In order to limit the usage of bought inputs like fossil fuels and agrochemicals, and to develop more diversified, resilient, and productive agroecosystems, they are more inclined to harness, sustain, and increase biological and ecological processes in the agricultural production. The model also shows that factors like grants, government policies, legislation, knowledge or local cohesion in clusters are important. At the same time, the model shows higher propensity to change among more informed farmers, able to exchange information with other framers or consultants in a systematic management, as with the innovation hubs established in this field. The lack of identification of a valuable business model constitutes an obstacle, easier to be overcome by organic farmers.

Early adopters are paying attention to diversification, mixed farming, intercropping, cultivar mixtures, habitat management methods for crop-associated biodiversity, biological pest control, bettering soil structure and health, biological nitrogen fixation, recycling of nutrients, energy, and waste [56].

Farmers in the marshes along the Danube River created the Bio Danubius Hub. Depending on the extent to which agroecological principles are followed locally, they are in the process of identifying categorized indicators on the inorganic-agroecological spectrum.

The results of this research indicated clearly that the proposed study is timely and pertinent in terms of the knowledge gap. Some of our findings are:

- reliance on ecological processes is not clear from the point of view of costs and benefits, and as a business model.

- local adaption and control of a system’s approach embracing management of interactions among components rather than focusing only on specific technologies is not clearly understood.

- understanding it as a social movement associated with agroecology is not obvious, and it requires greater efforts to initiate a widespread change of agriculture and food systems.

- agroecology is not understood as an interinstitutional political framework under which many social movements and peasant organizations around the world assert their collective rights, and advocate for a diversity of locally adapted agriculture and food systems, mainly practiced by small-scale food producers.

- there is a need for a strong connection to be made between agroecology, the right to food and food sovereignty, but it is not clear how to connect these areas.

- agroecology is seen as a political struggle, requiring people to challenge and transform governance structures and society at large, but it is not clear how and at what costs.

There are different challenges identified for the development of agroecology in Europe, as presented by Wezel et al. (2018) among which we list education and training, research funding, policies, consumer awareness, etc. [44]. Communication and the creation of alliances in the field of agroecology, through Innovation Hubs, contribute to the implementation of the concept starting from the local level, as a solid basis for support at the national and European level. A sustainable development must be done through education and innovation, and in this context, the Innovation Hubs are the core of the actions. Future activities that will be implemented in the short term through the Bio Danubius Hub include the introduction of the discipline of Agroecology in the curriculum of specialized faculties, such as economics and veterinary medicine; the creation of business models that support farmers so that work in agriculture becomes profitable; the elaboration of some legislative proposals in the field of agroecology that will be supported by the innovation hubs, as policy recommendations (stronger financial support for farmers, for example).

  1. Limitations of the study

Agriculture is the main scale at which the concept of agroecology is applied in Romania. In this context, the development of the conceptualization of agroecology in Romania takes place in parallel as a science and some agricultural practices and is just at the beginning, as a social movement. Innovation centers play an important role in its implementation. Correlation with the economic and political processes of the country is necessary, as emphasized by the farmers interviewed during the study. Future research related to this study can be related to the presentation of business models with impact in the practice of agroecology. Regionalization and nationalization of these concepts would also be necessary. Through innovation hubs, agroecology must move from practices at the scale of agroecosystems and agri-food systems, including economic, social, cultural and political aspects.

Re: The authors are grateful to all reviewers for the indications and considerations provided in their review. We feel that these indications have helped improving the quality of our paper. Thank you.

Reviewer 3 Report

Dear Authors,

the subject of the article is important and interesting.

However, I believe that the presented structure of article requires some changes. Comments below.

The Literature review section is too long and contains elements, that should rather be included in Discussion and conclusions section.

In Methodology chapter there is no information about the number of subjects covered by the research (what was the sample?). The hypotheses are not very original. This chapter also contains information (from line 390) that should be in the Results section.

In the Results chapter, the result are presented very generally.

The Discussion and Conclusions chapter is actually a summary and does not contain a scientific discussion.

In conclusion, the article can be published after changing its structure and confronting the research results with theory and other research carried out in Romania and other countries.

Author Response

  1. 3: Comments and Suggestions for Authors

Dear Authors,

the subject of the article is important and interesting. However, I believe that the presented structure of article requires some changes. Comments below.

  1. The Literature review section is too long and contains elements, that should rather be included in Discussion and conclusions section.

Authors’ answer: This section was eliminated from the article, and its content was redistributed in other sections of the article, according to their appropriateness, i.e., Sections 1, 2 and 5. All these changes are available in the Track changes version of the reviewed paper.

  1. In Methodology chapter there is no information about the number of subjects covered by the research (what was the sample?). The hypotheses are not very original. This chapter also contains information (from line 390) that should be in the Results section.

Authors’ answer: The new version of the article is grounded on specific literature review. I did not choose original hypotheses because there are no similar studies in Romania regarding ecological agriculture and in this way the processing method is important, which must be a safe one, certified by other authors.

The entire Methodology section has been rewritten. It now reads:

  1. Methodology

The current trend is the evolution of agroecology in Europe as well as in the world, as science, agricultural practice and social movement. In order to evaluate the degree of information and adoption of agroecology by farmers in Romania regarding agroecological practices, a questionnaire was offered for completion that included five sections: I. Personal, social, economic and demographic data; II. Identification of cultivation methods; III. Agroecological practices; IV. Identification and characterization of the conditions/factors related to the agricultural field. V. Importance and impact of innovation hubs in Romania. The questionnaire was answered online and aimed at identifying the knowledge of farmers regarding the methods of cultivation in an ecological system, the definition of sustainable agriculture, agricultural practices supporting biodiversity which are currently implemented, knowledge depth about the agroecology practices, on-farm specific problems, and types of regenerative agriculture practices.

The survey was anonymous, and it largely consisted of multiple-choice questions with Likert scale ratings (-2 not at all important; -1 not important; 0 neutral; 1 important; 2 extremely important). It included also open-ended questions allowing respondents to freely express their opinions. The outputs were evaluated using the statistical method of structural equation modeling using PLS, which examines concurrent interactions between latent variables, formative or reflective, even for smaller samples. This is preliminary research aiming at identifying suitable profiles for farmers and entrepreneurs in the agroecological field, to integrate them and offer specific support within innovation hubs. In this way, this exploratory study provides crucial data for further investigation. We designed our model based on 2 formative variables: Factors, Profile and one reflective variable Practices (Table 1 and Fig. 6).

The farmers that cultivated more that 5ha (Fig. 1), are paid for their activity or there are entrepreneurs in this field and know very well what the ecological agriculture is, they have a stronger entrepreneurial profile, are rather young men (26 - 45 years old) managing their own farm for less than 10 years (Fig. 2). They do not currently implement practices to support biodiversity, they accuse specific problems they have encountered on the farm (example) and do not want to change the farming system in their practice in the near       future, maybe because they learned how to practice agriculture from their family and      acquired their skills, knowledge and experience without attending specialized studies.

Fig.1. Agricultural area (ha) cultivated /leased          

Fig.2. Period of farm administration

/employment

Currently, they are only implementing biodiversity support practices to a small extent (Fig. 3.), (1-never, 5-very frequently) maybe because they have learned to practice farming from their family and acquired their skills, knowledge and experience without going through specialized studies, and they blame specific problems they encountered on the farm (for example, problems with human resources, lack of  employee skills, lack of technological resources, etc.).

         Fig. 3. Are you currently implementing practices that support biodiversity?

Fig.4. What do you understand by sustainable agriculture?

At the same time, it was important to know what the farmers understand by sustainable agriculture (Fig. 4.). A significant percentage of the respondents (46%) appreciated soil protection and biodiversity.

However, they are willing to change the farming system they currently practice, which can be a good start in supporting the transition to agroecology (Fig. 5).

Fig.5. Do you want to change the farming system you practice in the near future?

Table 1. Variables of the model

Variables

Items

Description

Profile

Education3

Regarding your education, please choose one of the options (which you have already graduated from)

Work4

Regarding your work: a) I am employed and paid, b) I am employed and entrepreneur, c) I am not paid, d) None of these

HA5

The agricultural area (ha) that you have cultivated / leased is: a) Less than 5ha, b) Between 5ha and 100ha c) Over 100ha

Time6

How long have you been managing your farm:

EcoAgri8

What do you mean by ecological agriculture?

SustenAgri9

What do you mean by sustainable agriculture?

ImplementP11

Are you currently implementing practices to support biodiversity?

Practices

Soil14

How do you want to change the farming system you practice in the near future? On a scale of 1 to 5 I want to add: [Ground cover]

Plowing14

How do you want to change the farming system you practice in the near future? On a scale of 1 to 5 I want to add: [Plowing]

Compost14

How do you want to change the farming system you practice in the near future? On a scale of 1 to 5 I want to add: [Compost, mulch, manure]

PestMng14

How do you want to change the farming system you practice in the near future? On a scale of 1 to 5 I want to add: [Integrated pest management]

Animal14

How do you want to change the farming system you practice in the near future? On a scale from 1 to 5 I want to add: [Integrated animal husbandry]

Culture14

How do you want to change the farming system you practice in the near future? On a scale of 1 to 5 I want to add: [Diversity of cultures]

Pollination14

How do you want to change the farming system you practice in the near future? On a scale of 1 to 5 I want to add: [Pollination]

Change13

Do you want to change the farming system you practice in the near future?

Factors

Subsidies20

Subsidies received

CostHa21

What are the costs per ha?

IncomeHa21

What are the incomes per ha?

Profit7

Does the farm offer you enough profits to live well?

Government 22

Are the government practices sustaining your activity?

Apreciaion26

How are you appreciated by your neighbors, in relation to the agricultural activity you carry out?

The hypotheses of the research are:

Quantitative research was based on closed questions and mostly on continuous categorial variables.

H1: The entrepreneurial profile of the Romanian farm manager is influenced by the conjunctural factors and thus organic farmers, and entrepreneurs are open /motivated for adoption of agroecological practices

H2: The entrepreneurial profile, their lever of literacy in the field of agroecology influences the practices of agroecology. We may affirm that there is a relevant gap of knowledge about these practices.

Qualitative research was based mostly on open questions

H3: Farmers organized in clusters are on a solid pathway towards innovation hubs

 Each variable of our model is composed by many items. For example, the formative variable Factors is made of six items evaluating the influence of the context on the entrepreneurial profile in agroecology filed. The item with the highest weight was CostHa21. The high loading factor (LF=0.700) of this variable emphasizes that the farmers consider that the cost per Ha is a very important factor that influences the agroecological practices.  Very related to cost is the profit. The entrepreneurs consider that the profit is the second important factor that influence these practices. The Profit17 variable has a loading factor of 0.413.   Other factors that influence the entrepreneurial profile are Susidies20 (LF=0.312), IncomeHa21 (LF=0.176), Appreciation26 (LF=0.120) and Gouvernment22 (0.097). Thus, we may affirm that the farmers are aware by the subsidies available in this field and use it in their activity. They are motivated by the fact that the neighbors appreciate the agricultural activity carried out by them and by the government support, in a small measure. (Fig. 6.).

The Profile variable is formed by 7 items and is rather positively influenced by the Time6 (LF=0.614), EcoAgri8 (LF=0.301), HA5 (LF=0.227), ImplementP11 (LF=0.245). Other factor that forms the user profile are Education (LF=0.137), SustenAgri9 (LF= 0.127) and Work4 (LF=0.082).  The farmers that work in the field by more than 5 years are keener to implement Ecological and Sustainable agriculture and sustain the biodiversity. The data show that some of them are entrepreneurs in this field and get profit from this activity (Fig. 6.).

The reflective variable Practices made of 8 items (Fig. 6) is rather influenced by the way of changing their farming system into the near future implementing Pwowing14 (LF=0.768), Animal14 (LF=0.691), Soil14 (LF=0.699), PestMng14 (LF=0.582), Pollination (LF=0.442), Compost14 (LF=0.343), Hedges14 (LF=0.178) and Changes13 (LF=0.474). Most of the farmers are already prepared to implement changes in their practice such as ground cover, plowing, compost, mulch, manure, Integrated pest management, Integrated animal husbandry, pollination and diversity of cultures.

  1. In the Results chapter, the result are presented very generally.

Authors’ answer: Our sample is not very big, but it is representative, because is based on a previous qualitative study, in which we already identified the main companies problems but we decided to have a quantitative expresion of our results.

The entire Results section has been rewritten.

  1. The Discussion and Conclusions chapter is actually a summary and does not contain a scientific discussion.

Authors’ answer: The entire Discussions, Conclusions section has been developed. In the Discussion section, the main results obtained were highlighted, especially those related to the entrepreneurial profile of the farm manager in Romania and the factors that influence it, the level of knowledge in the field of agroecology, the identification of some factors that motivate them in the approach to the transition to a sustainable agriculture , such as the costs, for example, short-term practices that can support the farmer and that can be implemented through legislative proposals at the national level.

At the same time, a separate section Limitations of the study was presented.

In conclusion, the article can be published after changing its structure and confronting the research results with theory and other research carried out in Romania and other countries.

Re: The authors are grateful to all reviewers for the indications and considerations provided in their review. We feel that these indications have helped improving the quality of our paper. Thank you.

  1. 3: Comments and Suggestions for Authors

Dear Authors,

the subject of the article is important and interesting. However, I believe that the presented structure of article requires some changes. Comments below.

  1. The Literature review section is too long and contains elements, that should rather be included in Discussion and conclusions section.

Authors’ answer: This section was eliminated from the article, and its content was redistributed in other sections of the article, according to their appropriateness, i.e., Sections 1, 2 and 5. All these changes are available in the Track changes version of the reviewed paper.

  1. In Methodology chapter there is no information about the number of subjects covered by the research (what was the sample?). The hypotheses are not very original. This chapter also contains information (from line 390) that should be in the Results section.

Authors’ answer: The new version of the article is grounded on specific literature review. I did not choose original hypotheses because there are no similar studies in Romania regarding ecological agriculture and in this way the processing method is important, which must be a safe one, certified by other authors.

The entire Methodology section has been rewritten. It now reads:

  1. Methodology

The current trend is the evolution of agroecology in Europe as well as in the world, as science, agricultural practice and social movement. In order to evaluate the degree of information and adoption of agroecology by farmers in Romania regarding agroecological practices, a questionnaire was offered for completion that included five sections: I. Personal, social, economic and demographic data; II. Identification of cultivation methods; III. Agroecological practices; IV. Identification and characterization of the conditions/factors related to the agricultural field. V. Importance and impact of innovation hubs in Romania. The questionnaire was answered online and aimed at identifying the knowledge of farmers regarding the methods of cultivation in an ecological system, the definition of sustainable agriculture, agricultural practices supporting biodiversity which are currently implemented, knowledge depth about the agroecology practices, on-farm specific problems, and types of regenerative agriculture practices.

The survey was anonymous, and it largely consisted of multiple-choice questions with Likert scale ratings (-2 not at all important; -1 not important; 0 neutral; 1 important; 2 extremely important). It included also open-ended questions allowing respondents to freely express their opinions. The outputs were evaluated using the statistical method of structural equation modeling using PLS, which examines concurrent interactions between latent variables, formative or reflective, even for smaller samples. This is preliminary research aiming at identifying suitable profiles for farmers and entrepreneurs in the agroecological field, to integrate them and offer specific support within innovation hubs. In this way, this exploratory study provides crucial data for further investigation. We designed our model based on 2 formative variables: Factors, Profile and one reflective variable Practices (Table 1 and Fig. 6).

The farmers that cultivated more that 5ha (Fig. 1), are paid for their activity or there are entrepreneurs in this field and know very well what the ecological agriculture is, they have a stronger entrepreneurial profile, are rather young men (26 - 45 years old) managing their own farm for less than 10 years (Fig. 2). They do not currently implement practices to support biodiversity, they accuse specific problems they have encountered on the farm (example) and do not want to change the farming system in their practice in the near       future, maybe because they learned how to practice agriculture from their family and      acquired their skills, knowledge and experience without attending specialized studies.

Fig.1. Agricultural area (ha) cultivated /leased          

Fig.2. Period of farm administration

/employment

Currently, they are only implementing biodiversity support practices to a small extent (Fig. 3.), (1-never, 5-very frequently) maybe because they have learned to practice farming from their family and acquired their skills, knowledge and experience without going through specialized studies, and they blame specific problems they encountered on the farm (for example, problems with human resources, lack of  employee skills, lack of technological resources, etc.).

         Fig. 3. Are you currently implementing practices that support biodiversity?

Fig.4. What do you understand by sustainable agriculture?

At the same time, it was important to know what the farmers understand by sustainable agriculture (Fig. 4.). A significant percentage of the respondents (46%) appreciated soil protection and biodiversity.

However, they are willing to change the farming system they currently practice, which can be a good start in supporting the transition to agroecology (Fig. 5).

Fig.5. Do you want to change the farming system you practice in the near future?

Table 1. Variables of the model

Variables

Items

Description

Profile

Education3

Regarding your education, please choose one of the options (which you have already graduated from)

Work4

Regarding your work: a) I am employed and paid, b) I am employed and entrepreneur, c) I am not paid, d) None of these

HA5

The agricultural area (ha) that you have cultivated / leased is: a) Less than 5ha, b) Between 5ha and 100ha c) Over 100ha

Time6

How long have you been managing your farm:

EcoAgri8

What do you mean by ecological agriculture?

SustenAgri9

What do you mean by sustainable agriculture?

ImplementP11

Are you currently implementing practices to support biodiversity?

Practices

Soil14

How do you want to change the farming system you practice in the near future? On a scale of 1 to 5 I want to add: [Ground cover]

Plowing14

How do you want to change the farming system you practice in the near future? On a scale of 1 to 5 I want to add: [Plowing]

Compost14

How do you want to change the farming system you practice in the near future? On a scale of 1 to 5 I want to add: [Compost, mulch, manure]

PestMng14

How do you want to change the farming system you practice in the near future? On a scale of 1 to 5 I want to add: [Integrated pest management]

Animal14

How do you want to change the farming system you practice in the near future? On a scale from 1 to 5 I want to add: [Integrated animal husbandry]

Culture14

How do you want to change the farming system you practice in the near future? On a scale of 1 to 5 I want to add: [Diversity of cultures]

Pollination14

How do you want to change the farming system you practice in the near future? On a scale of 1 to 5 I want to add: [Pollination]

Change13

Do you want to change the farming system you practice in the near future?

Factors

Subsidies20

Subsidies received

CostHa21

What are the costs per ha?

IncomeHa21

What are the incomes per ha?

Profit7

Does the farm offer you enough profits to live well?

Government 22

Are the government practices sustaining your activity?

Apreciaion26

How are you appreciated by your neighbors, in relation to the agricultural activity you carry out?

The hypotheses of the research are:

Quantitative research was based on closed questions and mostly on continuous categorial variables.

H1: The entrepreneurial profile of the Romanian farm manager is influenced by the conjunctural factors and thus organic farmers, and entrepreneurs are open /motivated for adoption of agroecological practices

H2: The entrepreneurial profile, their lever of literacy in the field of agroecology influences the practices of agroecology. We may affirm that there is a relevant gap of knowledge about these practices.

Qualitative research was based mostly on open questions

H3: Farmers organized in clusters are on a solid pathway towards innovation hubs

 Each variable of our model is composed by many items. For example, the formative variable Factors is made of six items evaluating the influence of the context on the entrepreneurial profile in agroecology filed. The item with the highest weight was CostHa21. The high loading factor (LF=0.700) of this variable emphasizes that the farmers consider that the cost per Ha is a very important factor that influences the agroecological practices.  Very related to cost is the profit. The entrepreneurs consider that the profit is the second important factor that influence these practices. The Profit17 variable has a loading factor of 0.413.   Other factors that influence the entrepreneurial profile are Susidies20 (LF=0.312), IncomeHa21 (LF=0.176), Appreciation26 (LF=0.120) and Gouvernment22 (0.097). Thus, we may affirm that the farmers are aware by the subsidies available in this field and use it in their activity. They are motivated by the fact that the neighbors appreciate the agricultural activity carried out by them and by the government support, in a small measure. (Fig. 6.).

The Profile variable is formed by 7 items and is rather positively influenced by the Time6 (LF=0.614), EcoAgri8 (LF=0.301), HA5 (LF=0.227), ImplementP11 (LF=0.245). Other factor that forms the user profile are Education (LF=0.137), SustenAgri9 (LF= 0.127) and Work4 (LF=0.082).  The farmers that work in the field by more than 5 years are keener to implement Ecological and Sustainable agriculture and sustain the biodiversity. The data show that some of them are entrepreneurs in this field and get profit from this activity (Fig. 6.).

The reflective variable Practices made of 8 items (Fig. 6) is rather influenced by the way of changing their farming system into the near future implementing Pwowing14 (LF=0.768), Animal14 (LF=0.691), Soil14 (LF=0.699), PestMng14 (LF=0.582), Pollination (LF=0.442), Compost14 (LF=0.343), Hedges14 (LF=0.178) and Changes13 (LF=0.474). Most of the farmers are already prepared to implement changes in their practice such as ground cover, plowing, compost, mulch, manure, Integrated pest management, Integrated animal husbandry, pollination and diversity of cultures.

  1. In the Results chapter, the result are presented very generally.

Authors’ answer: Our sample is not very big, but it is representative, because is based on a previous qualitative study, in which we already identified the main companies problems but we decided to have a quantitative expresion of our results.

The entire Results section has been rewritten.

  1. The Discussion and Conclusions chapter is actually a summary and does not contain a scientific discussion.

Authors’ answer: The entire Discussions, Conclusions section has been developed. In the Discussion section, the main results obtained were highlighted, especially those related to the entrepreneurial profile of the farm manager in Romania and the factors that influence it, the level of knowledge in the field of agroecology, the identification of some factors that motivate them in the approach to the transition to a sustainable agriculture , such as the costs, for example, short-term practices that can support the farmer and that can be implemented through legislative proposals at the national level.

At the same time, a separate section Limitations of the study was presented.

In conclusion, the article can be published after changing its structure and confronting the research results with theory and other research carried out in Romania and other countries.

Re: The authors are grateful to all reviewers for the indications and considerations provided in their review. We feel that these indications have helped improving the quality of our paper. Thank you.

Round 2

Reviewer 1 Report

The authors have addressed all of my comments. The paper can be accepted in current form

Author Response

Comments and Suggestions for Authors

The authors have addressed all of my comments. The paper can be accepted in current form

Re: Esteemed reviewer,

We are grateful that you have accepted our article for publication.

Warmest regards,

Laura Urdes

Reviewer 3 Report

Dear Authors,

chapter 3. Methodology still requires improvement.

In this chapter from verse 256 is a connection of methodology with research results that should not be there.

After correcting this chapter, the article can be published.

Author Response

Reviewer 3, Comments and Suggestions for Authors

Dear Authors,

chapter 3. Methodology still requires improvement.

In this chapter from verse 256 is a connection of methodology with research results that should not be there.

After correcting this chapter, the article can be published.

Authors’ reply: Thank you. Several connecting sections from the Methodology were revised and moved into the Results section. This change included line 256 (as indicated by the reviewer). The Results section was revised and some referenced authors in the References section and in text citations were updated accordingly. These changes can be viewed in the track changes version that is accompanying the note.

The Methodology section now reads:

  1. Methodology

The current trend is the evolution of agroecology in Europe as well as in the world, as science, agricultural practice and social movement. In order to evaluate the degree of information and adoption of agroecology by farmers in Romania regarding agroecological practices, a questionnaire was offered for completion that included five sections: I. Personal, social, economic and demographic data; II. Identification of culti-vation methods; III. Agroecological practices; IV. Identification and characterization of the conditions/factors related to the agricultural field. V. Importance and impact of in-novation hubs in Romania. The questionnaire was answered online and aimed at identifying the knowledge of farmers regarding the methods of cultivation in an ecological system, the definition of sustainable agriculture, agricultural practices supporting bio-diversity which are currently implemented, knowledge depth about the agroecology practices, on-farm specific problems, and types of regenerative agriculture practices.

The survey was anonymous, and it largely consisted of multiple-choice questions with Likert scale ratings (-2 not at all important; -1 not important; 0 neutral; 1 important; 2 extremely important). It included also open-ended questions allowing respondents to freely express their opinions. The outputs were evaluated using the statistical method of structural equation modeling using PLS, which examines concurrent interactions between latent variables, formative or reflective, even for smaller samples. This is preliminary research aiming at identifying suitable profiles for farmers and entrepreneurs in the agroecological field, to integrate them and offer specific support within innovation hubs. In this way, this exploratory study provides crucial data for further investigation. We designed our model based on 2 formative variables: Factors, Profile and one reflective variable Practices (Table 1).

Table 1. Variables of the model

Variables            Items    Description

Profile   Education3        Regarding your education, please choose one of the options (which you have al-ready graduated from)

              Work4  Regarding your work: a) I am employed and paid, b) I am employed and entrepre-neur, c) I am not paid, d) None of these

              HA5       The agricultural area (ha) that you have cultivated / leased is: a) Less than 5ha, b) Between 5ha and 100ha c) Over 100ha

              Time6   How long have you been managing your farm:

              EcoAgri8             What do you mean by ecological agriculture?

              SustenAgri9       What do you mean by sustainable agriculture?

              ImplementP11  Are you currently implementing practices to support biodiversity?

Practices             Soil14    How do you want to change the farming system you practice in the near future? On a scale of 1 to 5 I want to add: [Ground cover]

              Plowing14          How do you want to change the farming system you practice in the near future? On a scale of 1 to 5 I want to add: [Plowing]

              Compost14        How do you want to change the farming system you practice in the near future? On a scale of 1 to 5 I want to add: [Compost, mulch, manure]

              PestMng14         How do you want to change the farming system you practice in the near future? On a scale of 1 to 5 I want to add: [Integrated pest management]

              Animal14            How do you want to change the farming system you practice in the near future? On a scale from 1 to 5 I want to add: [Integrated animal husbandry]

              Culture14           How do you want to change the farming system you practice in the near future? On a scale of 1 to 5 I want to add: [Diversity of cultures]

              Pollination14     How do you want to change the farming system you practice in the near future? On a scale of 1 to 5 I want to add: [Pollination]

              Change13           Do you want to change the farming system you practice in the near future?

Factors Subsidies20       Subsidies received

              CostHa21           What are the costs per ha?

              IncomeHa21      What are the incomes per ha?

              Profit7  Does the farm offer you enough profits to live well?

              Government 22 Are the government practices sustaining your activity?

              Apreciaion26     How are you appreciated by your neighbors, in relation to the agricultural activity you carry out?

The hypotheses of the research are:

Quantitative research was based on closed questions and mostly on continuous categorial variables.

H1: The entrepreneurial profile of the Romanian farm manager is influenced by the conjunctural factors and thus organic farmers, and entrepreneurs are open /motivated for adoption of agroecological practices

H2: The entrepreneurial profile, their lever of literacy in the field of agroecology influences the practices of agroecology. We may affirm that there is a relevant gap of knowledge about these practices.

Qualitative research was based mostly on open questions

H3: Farmers organized in clusters are on a solid pathway towards innovation hubs

Taking into account the aforementioned hypotheses, the research employed SmartPls to assess the consistency through composite reliability.
